# Runtime Neural Pruning

**Ji Lin**[*]
Department of Automation
Tsinghua University
lin-j14@mails.tsinghua.edu.cn

**Yongming Rao**[*]
Department of Automation
Tsinghua University
raoyongming95@gmail.com

**Jiwen Lu**
Department of Automation
Tsinghua University
lujiwen@tsinghua.edu.cn

**Jie Zhou**
Department of Automation
Tsinghua University
jzhou@tsinghua.edu.cn

## Abstract

In this paper, we propose a Runtime Neural Pruning (RNP) framework which prunes the deep neural network dynamically at the runtime. Unlike existing neural pruning methods which produce a fixed pruned model for deployment, our method preserves the full ability of the original network and conducts pruning according to the input image and current feature maps adaptively. The pruning is performed in a bottom-up, layer-by-layer manner, which we model as a Markov decision process and use reinforcement learning for training. The agent judges the importance of each convolutional kernel and conducts channel-wise pruning conditioned on different samples, where the network is pruned more when the image is easier for the task. Since the ability of network is fully preserved, the balance point is easily adjustable according to the available resources. Our method can be applied to off-the-shelf network structures and reach a better tradeoff between speed and accuracy, especially with a large pruning rate.

## 1 Introduction

Deep neural networks have been proven to be effective in various areas. Despite the great success, the capability of deep neural networks comes at the cost of huge computational burdens and large power consumption, which is a big challenge for real-time deployments, especially for embedded systems. To address this, several neural pruning methods have been proposed [11, 12, 13, 25, 38] to reduce the parameters of convolutional networks, which achieve competitive or even slightly better performance. However, these works mainly focus on reducing the number of network weights, which have limited effects on speeding up the computation. More specifically, fully connected layers are proven to be more redundant and contribute more to the overall pruning rate, while convolutional layers are the most computationally dense part of the network. Moreover, such pruning strategy usually leads to an irregular network structure, *i.e.* with part of sparsity in convolution kernels, which needs a special algorithm for speeding up and is hard to harvest actual computational savings. A surprisingly effective approach to trade accuracy for the size and the speed is to simply reduce the number of channels in each convolutional layer. For example, Changpinyo *et al.* [27] proposed a method to speed up the network by deactivating connections between filters in convolutional layers, achieving a better tradeoff between the accuracy and the speed.

All these methods above prune the network in a fixed way, obtaining a static model for all the input images. However, it is obvious that some of the input sample are easier for recognition, which can be

---

[*]indicates equal contribution

recognized by simple and fast models. Some other samples are more difficult, which require more computational resources. This property is not exploited in previous neural pruning methods, where input samples are treated equally. Since some of the weights are lost during the pruning process, the network will lose the ability for some hard tasks forever. We argue that preserving the whole ability of the network and pruning the neural network dynamically according to the input image is desirable to achieve better speed and accuracy tradeoff compared to static pruning methods, which will also not harm the upper bound ability of the network.

In this paper, we propose a Runtime Neural Pruning (RNP) framework by pruning the neural network dynamically at the runtime. Different from existing methods that produce a fixed pruned model for deployment, our method preserves the full ability of the original network and prunes the neural network according to the input image and current feature maps. More specifically, we model the pruning of each convolutional layer as a Markov decision process (MDP), and train an agent with reinforcement learning to learn the best policy for pruning. Since the whole ability of the original network is preserved, the balance point can be easily adjusted according to the available resources, thus one single trained model can be adjusted for various devices from embedded systems to large data centers. Experimental results on the CIFAR [22] and ImageNet [36] datasets show that our framework successfully learns to allocate different amount of computational resources for different input images, and achieves much better performance at the same cost.

## 2   Related Work

**Network pruning:** There has been several works focusing on network pruning, which is a valid way to reduce the network complexity. For example, Hanson and Pratt [13] introduced hyperbolic and exponential biases to the pruning objective. Damage [25] and Surgeon [14] pruned the networks with second-order derivatives of the objective. Han *et al.* [11, 12] iteratively pruned near-zero weights to obtain a pruned network with no loss of accuracy. Some other works exploited more complicated regularizers. For example, [27, 44] introduced structured sparsity regularizers on the network weights, [32] put them to the hidden units. [17] pruned neurons based on the network output. Anwar *et al.* [2] considered channel-wise and kernel-wise sparsity, and proposed to use particle filters to decide the importance of connections and paths. Another aspect focuses on deactivating some subsets of connections inside a fixed network architecture. LeCun *et al.* [24] removed connections between the first two convolutional feature maps in a uniform manner. Depth multiplier method was proposed in [16] to reduce the number of filters in each convolutional layer by a factor in a uniform manner. These methods produced a static model for all the samples, failing to exploit the different property of input images. Moreover, most of them produced irregular network structures after pruning, which makes it hard to harvest actual computational savings directly.

**Deep reinforcement learning:** Reinforcement learning [29] aims to enable the agent to decide the behavior from its experiences. Unlike conventional machine learning methods, reinforcement learning is supervised through the reward signals of actions. Deep reinforcement learning [31] is a combination of deep learning and reinforcement learning, which has been widely used in recent years. For examples, Mnih *et al.* [31] combined reinforcement learning with CNN and achieved the human-level performance in the Atari game. Caicedo *et al.* [8] introduced reinforcement learning for active object localization. Zhang *et al.* [45] employed reinforcement learning for vision control in robotics. Reinforcement learning is also adopted for feature selection to build a fast classifier. [4, 15, 21].

**Dynamic network:** Dynamic network structures and executions have been studied in previous works [7, 28, 33, 39, 40]. Some input-dependent execution methods rely on a pre-defined strategy. Cascade methods [26, 28, 39, 40] relied on manually-selected thresholds to control execution. Dynamic Capacity Network [1] used a specially designed method to calculate a saliency map for control execution. Other conditional computation methods activate part of a network under a learned policy. Begio *et al.* [6] introduced Stochastic Times Smooth neurons as gaters for conditional computation within a deep neural network, producing a sparse binary gater to be computed as a function of the input. [5] selectively activated output of a fully-connected neural network, according to a control policy parametrized as the sigmoid of an affine transformation from last activation. Liu *et al.* [30] proposed Dynamic Deep Neural Networks (D2NN), a feed-forward deep neural network that allows selective execution with self-defined topology, where the control policy is learned using single step reinforcement learning.

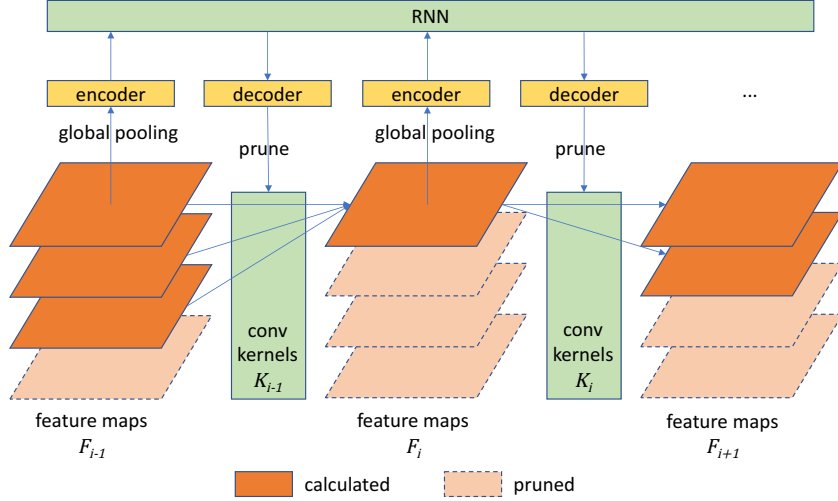

Figure 1: Overall framework of our RNP. RNP consists of two sub-networks: the backbone CNN network and the decision network. The convolution kernels of backbone CNN network are dynamically pruned according to the output Q-value of decision network, conditioned on the state forming from the last calculated feature maps.

## 3   Runtime Neural Pruning

The overall framework of our RNP is shown in Figure 1. RNP consists of two sub-networks, the backbone CNN network and the decision network which decides how to prune the convolution kernels conditioned on the input image and current feature maps. The backbone CNN network can be any kinds of CNN structure. Since convolutional layers are the most computationally dense layers in a CNN, we focus on the pruning of convolutional layers in this work, leaving fully connected layers as a classifier.

### 3.1   Bottom-up Runtime Pruning

We denote the backbone CNN with $m$ convolutional layers as $C$, with convolutional layers denoted as $C_1, C_2, ..., C_m$, whose kernels are $\mathbf{K}_1, \mathbf{K}_2, ..., \mathbf{K}_m$, respectively, with number of channels as $n_i, i = 1, 2, ..., m$. These convolutional layers produce feature maps $\mathbf{F}_1, \mathbf{F}_2, ..., \mathbf{F}_m$ as shown in Figure 1, with the size of $n_i \times H \times W, i = 1, 2, ..., m$. The goal is to find and prune the redundant convolutional kernels in $\mathbf{K}_{i+1}$, given feature maps $\mathbf{F}_i, i = 1, 2, ..., m - 1$, to reduce computation and achieve maximum performance simultaneously.

Taking the $i$-th layer as an example, we denote our goal as the following objective:

$$\min_{\mathbf{K}_{i+1}, h} \mathbb{E}_{\mathbf{F}_i}[L_{cls}(\text{conv}(\mathbf{F}_i, \mathbf{K}[h(\mathbf{F}_i)])) + L_{pnt}(h(\mathbf{F}_i))], \tag{1}$$

where $L_{cls}$ is the loss of the classification task, $L_{pnt}$ is the penalty term representing the tradeoff between the speed and the accuracy, $h(F_i)$ is the conditional pruning unit that produces a list of indexes of selected kernels according to input feature map, $K[\cdot]$ is the indexing operation for kernel pruning and $\text{conv}(x_1, x_2)$ is the convolutional operation for input feature map $x_1$ and kernel $x_2$. Note that our framework infers through standard convolutional layer after pruning, which can be easily boosted by utilizing GPU-accelerated neural network library such as cuDNN [9].

To solve the optimization problem in (1), we divide the whole problem into two sub-problems of $\{\mathbf{K}\}$ and $h$, and adopt an alternate training strategy to solve each sub-problem independently with the neural network optimizer such as RMSprop [42].

For an input sample, there are totally $m$ decisions of pruning to be made. A straightforward idea is using the optimized decisions under certain penalty to supervise the decision network. However, for a backbone CNN with $m$ layers, the time complexity of collecting the supervised signal is $\mathcal{O}(\prod_{i=1}^{m} n_m)$, which is NP-hard and unacceptable for prevalent very deep architecture such as

VGG [37] and ResNet [3]. To simplify the training problem, we employ the following two strategies: 1) model the network pruning as a Markov decision process (MDP) [34] and train the decision network by reinforcement learning; 2) redefine the action of pruning to reduce the number of decisions.

### 3.2 Layer-by-layer Markov Decision Process

The decision network consists of an encoder-RNN-decoder structure, where the encoder $E$ embeds feature map $\mathbf{F}_i$ into fixed-length code, RNN $R$ aggregates codes from previous stages, and the decoder $D$ outputs the Q-value of each action. We formulate key elements in Markov decision process (MDP) based on the decision network to adopt deep Q-learning in our RNP framework as follows.

**State:** Given feature map $\mathbf{F}_i$, we first extract a dense feature embedding $p_{\mathbf{F}_i}$ with global pooling, as commonly conducted in [10, 35], whose length is $n_i$. Since the number of channels for different convolutional layers are different, the length of $p_{\mathbf{F}_i}$ varies. To address this, we use the encoder $E$ (a fully connected layer) to project the pooled feature into a fixed-length embedding $E(p_{\mathbf{F}_i})$. $E(p_{\mathbf{F}_i})$ from different layers are associated in a bottom-up way with a RNN structure, which produces a latent code $R(E(p_{\mathbf{F}_i}))$, regarded as embedded state information for reinforcement learning. The decoder (also a fully connected layer) produces the Q-value for decision.

**Action:** The actions for each pruning are defined in an incremental way. For convolution kernel $\mathbf{K}_i$ with $n_i$ output channels, we determine which output channels are calculated and which to prune. To simplify the process, we group the output feature maps into $k$ sets, denoted as $\mathbf{F}'_1, \mathbf{F}'_2, ..., \mathbf{F}'_k$. One extreme case is $k = n_i$, where one single output channel forms a set. The actions $a_1, a_2, ..., a_k$ are defined as follows: taking actions $a_i$ yields calculating the feature map groups $\mathbf{F}'_1, \mathbf{F}'_2, ..., \mathbf{F}'_i$, $i = 1, 2, ..., k$. Hence the feature map groups with lower index are calculated more, and the higher indexed feature map groups are calculated only when the sample is difficult enough. Specially, the first feature map group is always calculated, which we mention as base feature map group. Since we do not have state information for the first convolutional layer, it is not pruned, with totally $m - 1$ actions to take.

Though the definitions of actions are rather simple, one can easily extend the definition for more complicated network structures. Like Inception [41] and ResNet [3], we define the action based on unit of a single block by sharing pruning rate inside the block, which is more scalable and can avoid considering about the sophisticated structures.

**Reward:** The reward of each action taken at the $t$-th step with action $a_i$ is defined as:

$$r_t(a_i) = \begin{cases} -\alpha L_{cls} + (i - 1) \times p, & \text{if inference terminates } (t = m - 1), \\ (i - 1) \times p, & \text{otherwise } (t < m - 1) \end{cases} \tag{2}$$

where $p$ is a negative penalty that can be manually set. The reward was set according to the loss for the original task. We took the negative loss $-\alpha L_{cls}$ as the final reward so that if a task is completed better, the final reward of the chain will be higher, *i.e.*, closer to 0. $\alpha$ is a hyper-parameter to rescale $L_{cls}$ into a proper range, since $L_{cls}$ varies a lot for different network structures and different tasks. Taking actions that calculate more feature maps, *i.e.*, with higher $i$, will bring higher penalty due to more computations. For $t = 1, ..., m - 2$, the reward is only about the computation penalty, while at the last step, the chain will get a final reward of $-\alpha L_{cls}$ to assess how well the pruned network completes the task.

The key step of the Markov decision model is to decide the best action at certain state. In other words, it is to find the optimal decision policy. By introducing the Q-learning method [31, 43], we define $Q(a_i, s_t)$ as the expectation value of taking action $a_i$ at state $s_t$. So the policy is defined as $\pi = \text{argmax}_{a_i} Q(a_i, s_t)$.

Therefore, the optimal action-value function can be written as:

$$Q(s_t, a_i) = \max_\pi \mathbb{E}[r_t + \gamma r_{t+1} + \gamma^2 r_{t+2} + ... | \pi], \tag{3}$$

where $\gamma$ is the discount factor in Q-learning, providing a tradeoff between the immediate reward and the prediction of future rewards. We use the decision network to approximate the expected Q-value $Q^*(s_t, a_i)$, with all the decoders sharing parameters and outputting a $k$-length vector, each representing the $Q^*$ of corresponding action. If the estimation is optimal, we will have $Q^*(s_t, a_i) = Q(s_t, a_i)$ exactly.

According to the Bellman equation [3], we adopt the squared mean error (MSE) as a criterion for training to keep decision network self-consistent. So we rewrite the objective for sub-problem of $h$ in optimization problem 1 as:

$$\min_{\theta} L_{re} = \mathbb{E}[r(s_t, a_i) + \gamma \max_{a_i} Q(s_{t+1}, a_i) - Q(s_t, a_i)]^2, \tag{4}$$

where $\theta$ is the weights of decision network. In our proposed framework, a series of states are created for an given input image. And the training is conducted using $\epsilon$-greedy strategy that selects actions following $\pi$ with probability $\epsilon$ and select random actions with probability $1 - \epsilon$, while inference is conducted greedily. The backbone CNN network and decision network is trained alternately. Algorithm 1 details the training procedure of the proposed method.

---

**Algorithm 1** Runtime neural pruning for solving optimization problem (1):

---

**Input:** training set with labels $\{X\}$
**Output:** backbone CNN $C$, decision network $D$
 1: **initialize:** train $C$ in normal way or initialize $C$ with pre-trained model
 2: **for** $i \leftarrow 1, 2, ..., M$ **do**
 3:     // train decision network
 4:     **for** $j \leftarrow 1, 2, ..., N_1$ **do**
 5:         Sample random minibatch from $\{X\}$
 6:         Forward and sample $\epsilon$-greedy actions $\{s_t, a_t\}$
 7:         Compute corresponding rewards $\{r_t\}$
 8:         Backward $Q$ values for each stage and generate $\nabla_{\theta} L_{re}$
 9:         Update $\theta$ using $\nabla_{\theta} L_{re}$
10:     **end for**
11:     // fine-tune backbone CNN
12:     **for** $k \leftarrow 1, 2, ..., N_2$ **do**
13:         Sample random minibatch from $\{X\}$
14:         Forward and calculate $L_{cls}$ after runtime pruning by $D$
15:         Backward and generate $\nabla_C L_{cls}$
16:         Update $C$ using $\nabla_C L_{cls}$
17:     **end for**
18: **end for**
19: **return** $C$ and $D$

---

It is worth noticing that during the training of agent, we manually set a fixed penalty for different actions and reach a balance status. While during deployment, we can adjust the penalty by compensating the output $Q^*$ of each action with relative penalties accordingly to switch between different balance point of accuracy and computation costs, since penalty is input-independent. Thus one single model can be deployed to different systems according to the available resources.

## 4 Experiments

We conducted experiments on three different datasets including CIFAR-10, CIFAR-100 [22] and ILSVRC2012 [36] to show the effectiveness of our method. For CIFAR-10, we used a four convolutional layer network with $3 \times 3$ kernels. For CIFAR-100 and ILSVRC2012, we used the VGG-16 network for evaluation. For results on the CIFAR dataset, we compared the results obtained by our RNP and naive channel reduction methods. For results on the ILSVRC2012 dataset, we compared the results achieved by our RNP with recent state-of-the-art network pruning methods.

### 4.1 Implementation Details

We trained RNP in an alternative manner, where the backbone CNN network and the decision network were trained iteratively. To help the training converge faster, we first initialized the CNN with random pruning, where decisions were randomly made. Then we fixed the CNN parameters and trained the decision network, regarding the backbone CNN as a environment, where the agent can take actions and get corresponding rewards. We fixed the decision network and fine-tuned the backbone CNN following the policy of the decision network, which helps CNN specialize in a specific task. The

initialization was trained using SGD, with an initial learning rate 0.01, decay by a factor of 10 after 120, 160 epochs, with totally 200 epochs in total. The other training progress was conducted using RMSprop [42] with the learning rate of 1e-6. For the $\epsilon$-greedy strategy, the hyper-parameter $\epsilon$ was annealed linearly from 1.0 to 0.1 in the beginning and fixed at 0.1 thereafter.

For most experiments, we set the number of convolutional group to $k = 4$, which is a tradeoff between the performance and the complicity. Increasing $k$ will enable more possible pruning combinations, while at the same time making it harder for reinforcement learning with an enlarged action space. Since the action is taken conditioned on the current feature map, the first convolutional layer is not pruned, where we have totally $m - 1$ decisions to make, forming a decision sequence. During the training, we set the penalty for extra feature map calculation as $p = -0.1$, which is adjusted during the deployment. The scale $\alpha$ factor was set such that the average $\alpha L_{cls}$ is approximately 1 to make the relative difference more significant. For experiments on VGG-16 model, we define the actions based on unit of a single block by sharing pruning rate inside the block as mentioned in Section 3.2 to simplify implementation and accelerate convergence.

For vanilla baseline methods comparison on CIFAR, we evaluated the performance of normal neural network with the same computations. More specifically, we calculated the average number of multiplications of every convolution layer and rounded it up to the nearest number of channels sharing same computations, which resulted in an identical network topology with reduced convolutional channels. We trained the vanilla baseline network with the SGD until convergence for comparison. All our experiments were implemented using the modified Caffe toolbox [20].

## 4.2 Intuitive Experiments

To have an intuitive understanding of our framework, we first conducted a simple experiment to show the effectiveness and undergoing logic of our RNP. We considered a 3-category classification problem, consisting of male faces, female faces and background samples. It is intuitive to think that separating male faces from female faces is a much more difficult task than separating faces from background, needing more detailed attention, so more resources should be allocated to face images than background images. In other words, a good tradeoff for RNP is to prune the neural network more when dealing with background images and keep more convolutional channels when inputting a face image.

To validate this idea, we constructed a 3-category dataset using Labeled Faces in the Wild [18] dataset, which we referred to as LFW-T. More specifically, we randomly cropped 3000 images for both male and female faces, and also 3000 background images randomly cropped from LFW. We used the attributes from [23] as labels for male and female faces. All these images were resized to $32 \times 32$ pixels. We held out 2000 images for testing and the remaining for training. For this experiment, we designed a 3-layer convolutional network with two fully connected layers. All convolutional kernels are $3 \times 3$ and with 32, 32, 64 output channels respectively. We followed the same training protocol as mentioned above with $p = 0.1$, and focused on the difference between different classes.

The original network achieved 91.1% accuracy. By adjusting the penalty, we managed to get a certain point of accuracy-computation tradeoff, where computations (multiplications) were reduced by a factor of 2, while obtaining even slightly higher accuracy of 91.75%. We looked into the average computations of different classes by counting multiplications of convolutional layers. The results were shown in Figure 2. For the whole network, RNP allocated more computations on faces images than background images, at approximately a ratio of 2, which clearly demonstrates the effectiveness of RNP. However, since the first convolutional layers and fully connected layers were not pruned, to get the absolute ratio of pruning rate, we also studied the pruning of a certain convolutional layer. In this case, we selected the last convolutional layer `conv3`. The results are shown on the right figure. We see that for this certain layer, computations for face images are almost 5 times of background images. The differences in computations show that RNP is able to find the relative difficulty of different tasks and exploit such property to prune the neural network accordingly.

## 4.3 Results

**CIFAR-10 & CIFAR-100:** For CIFAR-10 and CIFAR-100, we used a four-layer convolutional network and the VGG-16 network for experiments, respectively. The goal of these two experiments is to compare our RNP with vanilla baseline network, where the number of convolutional layers was

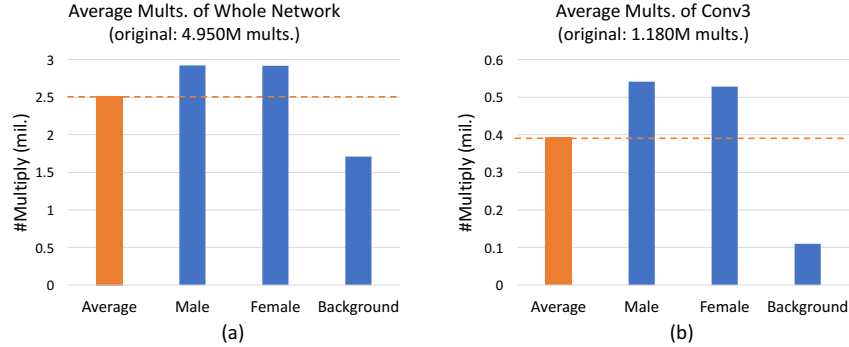

Figure 2: The average multiplication numbers of different classes in our intuitive experiment. We show the computation numbers for both the whole network (on the left) and the fully pruned convolutional layer conv3 (on the right). The results show that RNP succeeds to focus more on faces images by preserving more convolutional channels while prunes the network more when dealing with background images, reaching a good tradeoff between accuracy and speed.

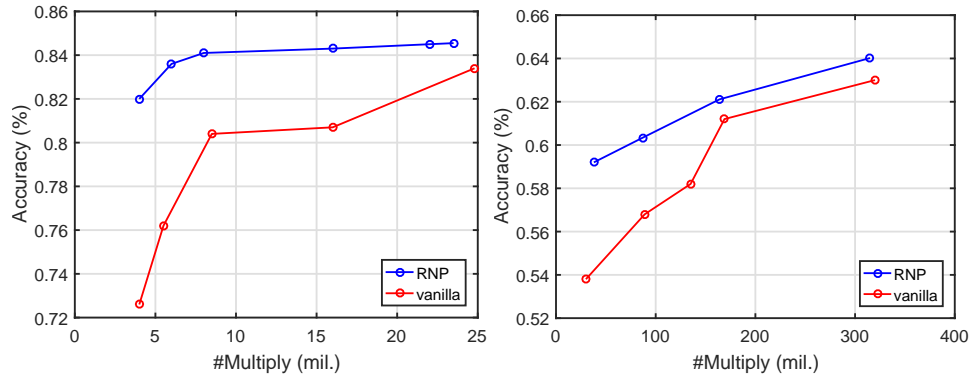

Figure 3: The results on CIFAR-10 (on the left) and CIFAR-100 (on the right). For vanilla curve, the rightmost point is the full model and the leftmost is the $\frac{1}{4}$ model. RNP outperforms naive channel reduction models consistently by a very large margin.

reduced directly from the beginning. The fully connected layers of standard VGG-16 are too redundant for CIFAR-100, so we eliminated one of the fully connected layer and set the inner dimension as 512. The modified VGG-16 model was easier to converge and actually slightly outperformed the original model on CIFAR-100. The results are shown in Figure 3. We see that for vanilla baseline method, the accuracy suffered from a stiff drop when computations savings were than 2.5 times. While our RNP consistently outperformed the baseline model, and achieved competitive performance even with a very large computation saving rate.

**ILSVRC2012:** We compared our RNP with recent state-of-the-art neural pruning methods [19, 27, 46] on the ImageNet dataset using the VGG-16 model, which won the 2-nd place in ILSVRC2014 challenge. We evaluated the *top-5 error* using single-view testing on ILSVRC2012-val set and trained RNP model using ILSVRC2012-train set. The view was the center $224 \times 224$ region cropped from the

Table 1: Comparisons of increase of top-5 error on ILSVRC2012-val (%) with recent state-of-the-art methods, where we used $10.1\%$ top-5 error baseline as the reference.

| Speed-up | $3\times$ | $4\times$ | $5\times$ | $10\times$ |
|---|---|---|---|---|
| Jaderberg *et al.* [19] ([46]'s implementation) | **2.3** | 9.7 | 29.7 | |
| Asymmetric [46] | - | 3.84 | - | |
| Filter pruning [27] (our implementation) | 3.2 | 8.6 | 14.6 | |
| **Ours** | 2.32 | **3.23** | **3.58** | **4.89** |

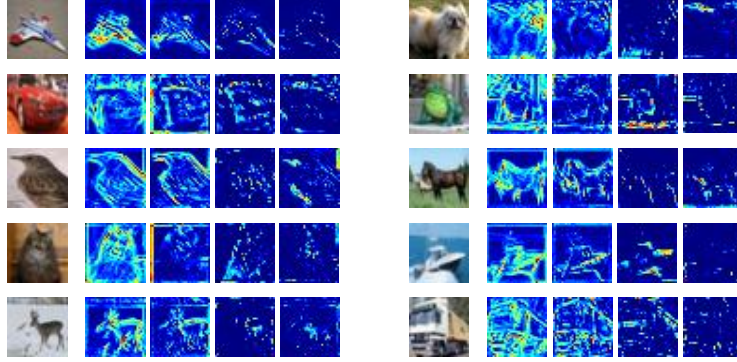

Figure 4: Visualization of the original images and the feature maps of four convolutional groups, respectively. The presented feature maps are the average of corresponding convolutional groups.

Table 2: GPU inference time under different theoretical speed-up ratios on ILSVRC2012-val set.

| Speed-up solution | Increase of top-5 error (%) | Mean inference time (ms) |
|---|---|---|
| VGG-16 ($1\times$) | 0 | 3.26 ($1.0\times$) |
| Ours ($3\times$) | 2.32 | 1.38 ($2.3\times$) |
| Ours ($4\times$) | 3.23 | 1.07 ($3.0\times$) |
| Ours ($5\times$) | 3.58 | 0.880 ($3.7\times$) |
| Ours ($10\times$) | 4.89 | 0.554 ($5.9\times$) |

resized images whose shorter side is 256 by following [46]. RNP was fine-tuned based on the public available model [2] which achieves $10.1\%$ top-5 error on ILSVRC2012-val set. Results are shown in Table 1, where *speed-up* is the theoretical speed-up ratio computed by the complexity. We see that RNP achieves similar performance with a relatively small speed-up ratio with other methods and outperforms other methods by a significant margin with a large speed-up ratio. We further conducted our experiments on larger ratio ($10\times$) and found RNP only suffered slight drops ($1.31\%$ compared to $5\times$), far beyond others' results on $5\times$ setting.

## 4.4 Analysis

**Analysis of Feature Maps:** Since we define the actions in an incremental way, the convolutional channels of lower index are calculated more (a special case is the base network that is always calculated). The convolutional groups with higher index are increments to the lower-indexed ones, so the functions of different convolution groups might be similar to "low-frequency" and "high-frequency" filters. We visualized different functions of convolutional groups by calculating average feature maps produced by each convolutional group. Specially, we took CIFAR-10 as example and visualized the feature maps of `conv2` with $k = 4$. The results are shown in Figure 4.

From the figure, we see that the base convolutional groups have highest activations to the input images, which can well describe the overall appearance of the object. While convolutional groups with higher index have sparse activations, which can be considered as a compensation to the base convolutional groups. So the undergoing logic of RNP is to judge when it is necessary to compensate the base convolutional groups with higher ones: if tasks are easy, RNP will prune the high-order feature maps for speed, otherwise bring in more computations to pursue accuracy.

**Runtime Analysis:** One advantage of our RNP is its convenience for deployment, which makes it easy to harvest actual computational time savings. Therefore, we measured the actual runtime under GPU acceleration, where we measured the actual inference time for VGG-16 on ILSVRC2012-val set. Inference time were measured on a Titan X (Pascal) GPU with batch size 64. Table 2 shows the GPU inference time of different settings. We see that our RNP generalizes well on GPU.

## 5    Conclusion

In this paper, we have proposed a Runtime Neural Pruning (RNP) framework to prune the neural network dynamically. Since the ability of network is fully preserved, the balance point is easily adjustable according to the available resources. Our method can be applied to off-the-shelf network structures and reaches a better tradeoff between speed and accuracy. Experimental results demonstrated the effectiveness of the proposed approach.

## Acknowledgements

We would like to thank Song Han, Huazhe (Harry) Xu, Xiangyu Zhang and Jian Sun for their generous help and insightful advice. This work is supported by the National Natural Science Foundation of China under Grants 61672306 and the National 1000 Young Talents Plan Program. The corresponding author of this work is Jiwen Lu.

## Footnotes

[2] http://www.robots.ox.ac.uk/~vgg/research/very_deep/

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
