[Reviews · NeurIPS 2017]

Reviewer 1



The paper proposes a run-time pruning algorithm for deep networks where pruning is performed according to the input image and features. RL is used to judge the importance of each convolutional kernel and conduct channel-wise pruning. The method can be applied to off-the-shelf networks and a tradeoff between speed and accuracy can be achieved. Positives - The idea of doing dynamic pruning of neural networks by learning the decision to prune using RL is fairly novel. - The idea is justified well using empirical evaluation on Imagenet and CIFAR datasets with relevant comparisons to other pruning methods and trade-off between speed and accuracy.

Reviewer 2



**summary: This paper proposes to prune convolutional layers of CNN dynamically at runtime. The authors formulate the pruning as a MDP. Specifically, at each layer the feature map is embedded as a fixed length code. A decoder then produces the Q-value for decision. The actions correspond to which group of channels to compute and which to prune and the reward relates to the cost as well as the loss of the prediction. Experiments on CIFAR10/100 and ImageNet show improvement over other CNN pruning techniques. **other comments: The paper applies the ideas of deep reinforcement learning to CNN pruning. The procedure is clearly presented and the intuition well-explained. The idea of using MDP techniques to solve resource-constrained classification is not new. The authors may want to cite the paper by Benbouzid et al 2012: Fast classification using sparse decision DAGs. A potential drawback of the presented method is its complexity. To train a deep Q network together with a CNN seems highly non-trivial. In practice, simpler network compression methods might be preferred. Have the authors compared to other model compression methods such as in Quantized Convolutional Neural Networks for Mobile Device by Wu et al 2016? **overall: the paper is well developed and novel in its application area. It's a good paper if it has thoroughly compared to recent model compression papers for CNN to justify its added complexity.

Reviewer 3



Update: I updated my score after RNN overhead clarification for more recent networks and their remark on training complexity. The authors propose a deep RL based method to choose a subset of convolutional kernels in runtime leading to faster evaluation speed for CNNs. I really like the idea of combining an RNN and using it to guide network structure. I have some doubts on the overhead of the decision network (see below) would like a comment on that before making my final decision. -Do your computation results include RNN runtime (decision network), can you comment on the overhead? The experiments focus on VGG16 which is a very heavy network. In comparison, more efficient networks like GoogleNet may make the RNN overhead significant. Do you have time or flops measurements for system overhead? Finally, as depth increases (e.g. Resnet151) your decision overhead increases as well since the decider network runs at every step. - Do you train a different RNP for each p? Is 0.1 a magic number? -The reward structure and formulating feature (in this case kernel) selection as an MDP is not new, check Timely Object Recognition Karayev, Imitation Learning by Coaching, Hal Daume III. However, application of it to internal kernel selection is interesting. line 33 "Some other samples are more difficult, which require more computational resources. This property is not exploited in previous neural pruning methods, where input samples are treated equally" This is not exactly true check missing references below. Missing references on sample adaptive inference in DNNs: -The cascading neural network: building the Internet of Smart Things Leroux et al. -Adaptive Neural Networks for Fast Test-Time Prediction Bolukbasi et al. -Changing Model Behavior at Test-Time Using Reinforcement Learning Odena et al. misc: table 1 3x column should have Jaderberg as bold (2.3 vs 2.32) line 158 feature -> future line 198 p=-0.1 -> p=0.1 ?